# Trappc9 Deficiency Impairs the Plasticity of Stem Cells

**DOI:** 10.3390/ijms23094900

**Published:** 2022-04-28

**Authors:** Muhammad Usman, Yan Li, Yuting Ke, Gaurav Chhetri, Md Ariful Islam, Zejian Wang, Xueyi Li

**Affiliations:** 1School of Pharmacy, Shanghai Jiao Tong University, 800 Dong Chuan Road, Shanghai 200240, China; pharmacistusman@sjtu.edu.cn (M.U.); 018170210001@sjtu.edu.cn (Y.L.); yke@mit.edu (Y.K.); gvchhetri@sjtu.edu.cn (G.C.); mdarifulislam@sjtu.edu.cn (M.A.I.); wangzejian@sjtu.edu.cn (Z.W.); 2Department of Neurology, Massachusetts General Hospital and Harvard Medical School, Charlestown, MA 02129, USA

**Keywords:** obesity, adipose stem cells, cellular senescence, trappc9, rab18

## Abstract

Genetic mutations of *trappc9* cause intellectual disability with the atrophy of brain structures and variable obesity by poorly understood mechanisms. Trappc9-deficient mice develop phenotypes resembling pathological changes in humans and appear overweight shortly after weaning, and thus are useful for studying the pathogenesis of obesity. Here, we investigated the effects of trappc9 deficiency on the proliferation and differentiation capacity of adipose-derived stem cells (ASCs). We isolated ASCs from mice before overweight was developed and found that trappc9-null ASCs exhibited signs of premature senescence and cell death. While the lineage commitment was retained, trappc9-null ASCs preferred adipogenic differentiation. We observed a profound accumulation of lipid droplets in adipogenic cells derived from trappc9-deficient ASCs and marked differences in the distribution patterns and levels of calcium deposited in osteoblasts obtained from trappc9-null ASCs. Biochemical studies revealed that trappc9 deficiency resulted in an upregulated expression of rab1, rab11, and rab18, and agitated autophagy in ASCs. Moreover, we found that the content of neural stem cells in both the subventricular zone of the lateral ventricle and the subgranular zone of the dentate gyrus vastly declined in trappc9-null mice. Collectively, our results suggest that obesity, as well as brain structure hypoplasia induced by the deficiency of trappc9, involves an impairment in the plasticity of stem cells.

## 1. Introduction

Obesity is a chronic disease characterized by the excessive accumulation of body fat and has been becoming a serious global health problem as a major risk factor for a variety of severe medical conditions, including type-II diabetes mellitus, heart disease, stroke, and certain types of cancer [1,2]. The prevalence of obesity continues to increase in both children and adults; by 2030, half of the global population is expected to be overweight or obese [2]. While it is often associated with an imbalance between calorie intake and energy expenditure, the etiology of obesity is complex and involves genetic, environmental, physiological, psychological, economic, and even political factors [2].

Mesenchymal stem cells (MSCs) are non-hematopoietic stromal cells capable of differentiation into multiple mesodermal cell types, including adipocytes, chondrocytes, osteocytes, and myocytes [3,4]. Adipose tissue-resident MSCs, often referred to as adipose-derived stem or stromal cells (ASCs), are cardinal regulators of adipose tissue remodeling by giving rise to adipocytes and maintaining metabolic and immunologic homeostasis in adipose tissues [5,6,7]. Increasing evidence suggests a feed-forward vicious cycle between obesity and MSC dysfunction in the development of obesity [8]. In this scenario, obesity-associated low-grade systemic inflammation and cellular changes, e.g., mitochondrial dysfunction and epigenetic reprogramming, render MSC dysfunction, which in turn accelerates the development of obesity and co-morbidities.

Trappc9 is a subunit of the highly conserved protein complex called the transport protein particle (TRAPP), a guanine nucleotide exchange factor for rab proteins that operate in secretory, endocytic, and autophagic pathways [9]. Genetic mutations causing trappc9 loss-of-function have been linked to intellectual disability, autism spectrum disorder, schizophrenia, and attention deficit hyperactivity disorder [10,11,12,13,14,15,16]. Over half of the patients with trappc9 mutations are reported to present different degrees of obesity [17], suggesting that trappc9 has a role in the development of obesity. A recent study showed that patients with trappc9 mutations displayed global metabolic changes detectable in the peripheral blood [17]. Yet, how trappc9 loss induces obesity is not clear.

We have generated a line of mice deficient for trappc9, which develops phenotypes resembling pathologies in human [18]. Compared with their wildtype (WT) or heterozygous littermates, trappc9-null mice are normal in growth before weaning, but exhibit an increase in gaining body weight shortly after weaning [18]. This feature makes this mouse line a useful tool for studying obesity. In this study, we investigated the effects of trappc9 deficiency on the plasticity of stem cells. We found that ASCs from trappc9-null mice exhibited signs of increased stress and apoptosis and were altered in the differentiation to both adipogenic and osteogenic lineages when compared with ASCs from WT mice. Biochemical studies revealed that multiple cellular pathways were perturbed in trappc9-null ASCs. Our study suggests that trappc9 deficiency impinges ASCs, thereby promoting the development of obesity.

## 2. Results

### 2.1. Loss of Trappc9 Restricts the Proliferation of ASCs

To examine effects of trappc9 deficiency on the plasticity of stem cells, we prepared ASCs from the abdominal fat pads of trappc9-null mice and their WT littermates. We used 3-week-old mice for such preparations to minimize the adverse effects of obesity on ASCs, because trappc9-null mice at this age had not developed overweight or obesity yet [18] (Appendix A). As expected, ASCs isolated from trappc9-null and WT mice were positive for MSC-specific CD29, CD44, and Sca1 (Stem cell antigen 1) and negative for hematopoietic CD45 (Appendix A and Figure 1A). WT and trappc9-null ASCs were adherent and exhibited a fibroblast-like morphology during growth on culture dishes (Figure 1A and Appendix A). However, trappc9-null ASCs appeared larger and grew more slowly than WT ASCs (Figure 1B and Appendix A). An analysis of immunofluorescence microscopic images revealed a greater population of cells containing two or more nuclei in trappc9-null ASC cultures than in WT ASC cultures (Figure 1C). We noticed loss of cells in trappc9-deficient ASC cultures at the early stages of culture (Appendix A). Under the same conditions, cell loss was barely seen in WT ASC cultures. Cell loss in trappc9-null ASC cultures might arise from cell migration to other areas and/or from cell death.

To determine whether trappc9 deficiency predisposes ASCs to apoptosis, we conducted flow cytometry analysis to detect phosphatidylserine exposed outside of cells with fluorescein-labeled Annexin-V and to determine the integrity of plasma membranes with propidium iodide. While there was no significant difference in the population of cells at the late stages of apoptosis (positive for both Annexin-V and propidium iodide) between WT and trappc9-deficient ASCs, the population of cells at the early stages of apoptosis (Annexin-V positive but propidium iodide negative) was significantly increased in trappc9-deficient ASC cultures compared with that in WT ASC cultures (Figure 1D; *n* = 3, Mean ± SD percentage, WT vs. trappc9-null, Student’s *t*-test, early apoptosis: 8.81 ± 0.995 vs. 22.3 ± 1.345, *p* < 0.01; late apoptosis: 5.05 ± 0.136 vs. 8.73 ± 2.659, no significance). On the contrary, the viable (negative for both Annexin-V and propidium iodide) population of cells in trappc9-null ASC cultures declined relative to WT ASC cultures (Figure 1D; *n* = 3, Mean ± SD percentage, WT vs. trappc9-null, Student’s *t*-test, 85.26 ± 1.078 vs. 68.56 ± 1.855, *p* < 0.001). Taken together, our data suggest that the deficiency of trappc9 constrains the proliferation of ASCs.

### 2.2. Trappc9 Deficiency Skews ASCs to Adipogenic Differentiation

To determine whether a loss of trappc9 perturbs the differentiation capacity of ASCs, WT and trappc9-deficient ASCs at passage-4 were induced to differentiate to the adipogenic or osteogenic lineages. Within 14 days of induction in vitro, both WT and trappc9-deficient ASCs successfully generated cells stainable with Oil red O or with Alizarin red (Figure 2A,B), suggesting that trappc9-deficient ASCs still possess the capability of multi-lineage differentiation. However, quantitative analysis revealed that trappc9-null ASCs generated a higher proportion of oil red stained cells and a lower proportion of Alizarin red stained cells when compared with WT ASCs (Figure 2C,D), indicating that trappc9 deficiency in ASCs disturbs the balance of differentiation between the adipogenic and osteogenic lineages. In observance that after induced differentiation osteoblast cells were too dense to count accurately, we used the formation of bone-like calcified nodules as a measure for comparing the osteogenic differentiation between WT and trappc9-deficient ASCs, as this was widely used previously [19]. Under the same induction conditions, calcified nodules were less abundantly formed in osteogenic cells derived from trappc9-deficient ASCs than from WT ASCs (Figure 2D). To further demonstrate the biased differentiation of trappc9-null ASCs, we examined the levels of transcripts of adipogenic or osteogenic lineage genes. A quantitative RT-PCR analysis showed an upregulation of adipocyte-related transcripts (*Ap2*, *Lpl*, and *Pparg*) and a downregulation of osteoblast-specific transcripts (*Bsp*, *Runx2*, and Oc) in trappc9-null ASCs relative to their abundance in WT ASCs (Figure 2E,F). Collectively, these data suggest that trappc9 loss in ASCs does not impair the lineage commitment, but skews their fate of differentiation.

While trappc9-deficient ASCs were able to differentiate into cells of different lineages, adipogenic and osteogenic cells derived from trappc9-deficient ASCs were markedly different from their counterparts obtained from WT ASCs. Adipogenic cells from trappc9-deficient ASCs had a greater content of lipids than those derived from WT ASCs, as indicated by the signal intensities of Oil red O staining (Figure 2A,G). Furthermore, aberrantly large lipid droplets were frequently seen in adipogenic cells derived from trappc9-null ASCs, but were rare in those obtained from WT ASCs (Figure 2A). On the other hand, osteoblasts obtained from trappc9-null ASCs were less abundantly enriched in calcium than those derived from WT ASCs (Figure 2H). Additionally, calcium deposits in WT ASC-derived osteoblasts were distributed as a cytoskeleton-like network, whereas those in osteoblasts from trappc9-null ASCs were irregularly distributed, with the string-like structures being kinked or shortened (Figure 2B). These data suggest that trappc9 loss in ASCs disorganizes cellular compartments in their relevant progeny cells.

### 2.3. Trappc9-Null ASCs Are Prone to Premature Senescence

In observance that trappc9-null ASCs at early passages present signs of senescence, more specifically an enlarged and flattened appearance, poor proliferation, and an unbalanced lineage differentiation, we speculated that trappc9 deficiency accelerated ASCs to enter the senescent state. Indeed, staining for senescence-associated beta galactosidase revealed more positive cells in trappc9-null ASC cultures at passage-4 than in passage-matched WT ASC cultures (Figure 3A,B). Furthermore, qPCR analysis showed that senescence-associated genes (*p16*, *p21*, and *p53*) were expressed at higher levels in trappc9-null ASCs than in WT ASCs (Figure 3C). These data suggest that the deficiency of trappc9 predisposes ASCs to premature senescence.

Oxidative stress is a major trigger of cellular senescence [20,21]. To determine whether oxidative stress occurs in trappc9-deficient ASCs at early passages, we compared levels of intracellular reactive oxygen species (ROS) between trappc9-null and WT ASCs. To this end, trappc9-null and WT ASCs at passage-4 were treated with 5-(and-6)-carboxy-2′,7′-dichlorofluorescein diacetate (DCFDA), a cell-permeable non-fluorescent dye that is oxidized into the highly fluorescent DCF and widely used for measuring levels of ROS in cells [22]. Consistent with the notion that low levels of ROS are necessary for physiologic functions of stem cells [23], DCF signals were detected in both WT and trappc9-null ASCs (Figure 3D). However, levels of DCF signals in trappc9-null ASCs were significantly higher than those in WT ASCs (Figure 3D,E), suggesting that there is a bona fide accumulation of excessive ROS in trappc9-deficient ASCs.

### 2.4. Derangements of Cellular Trafficking Pathways in Trappc9-Deficient ASCs

Having found the dysfunction and premature senescence of trappc9-null ASCs, we then looked for molecular changes rendered by the loss of trappc9 in ASCs. Trappc9 is a subunit specific for TRAPP-II, which has been shown to be an activator for rab1, rab11, and rab18 [18,24,25]. The abrogation of trappc9 in mice elevates the expression levels of rab1 and rab11 but not rab18 in the brain [18]. We conducted a Western blot analysis to compare the expression levels of these rabs in WT and trappc9 ASCs. Consistent with their upregulated expression in the brain of trappc9-null mice, levels of rab1 and rab11 were higher in trappc9-null ASCs than in WT ASCs (Figure 4A,B). Of note, rab18, which is not altered in the brain of trappc9-null mice [18], was also upregulated in trappc9-deficient ASCs (Figure 4A,B), suggesting that trappc9 loss may render a cell-type or tissue-specific impact on rab18. The elevated expression of rab1, rab11, and rab18 in trappc9-deficient ASCs was unlikely to be a nonspecific effect because under the same conditions, the expression levels of rab5 and rab7 in trappc9-null ASCs were similar to those in WT ASCs. Taken together, these data suggest that a loss of trappc9 in ASCs may alter pathways involving rab1, rab11, and/or rab18.

As rab1, rab11, and rab18 all have a regulatory role in autophagy, which has been shown to be involved in cellular senescence [26,27,28,29,30,31,32,33], we examined the expression levels of autophagy-related proteins to determine whether autophagy was altered in trappc9-null ASCs. Western blot analysis revealed that Beclin-1, LC3, and p62/SQSTM1 were all expressed at higher levels in trappc9-null ASCs than in WT ASCs (Figure 4C,D). Upon autophagy activation, cytoplasmic LC3 (LC3-I) is converted to its lipid-conjugated form LC3-II, which localizes to the membranes of autophagosomes; the ratio of LC3-II to LC3-I is thought to reflect the amplitude of autophagy activation [34]. We calculated the LC3-II to LC3-I ratio and found that it was vastly elevated in trappc9-null ASCs relative to that in WT ASCs (Figure 4C,D), suggesting that autophagy is enhanced in trappc9-deficient ASCs. However, the autophagic flux to lysosomes for degradation in trappc9-null ASCs appeared to decline, as indicated by the accumulation or increased levels of autophagy substrates p62/SQSTM1, as well as LC3-II. Collectively, these data suggest that trappc9 loss enhances autophagy but reduces autophagic degradation in ASCs.

Rab18 also plays a crucial role in the dynamics of lipid droplets, whose accumulation is a hallmark of obesity [35,36,37]. Our above studies showed that adipogenic cells derived from trappc9-null ASCs had an accumulation of aberrantly large lipid droplets, which were rarely seen in adipogenic cells derived from WT ASCs (Figure 2A). As enlarged lipid droplets were already seen in un-induced trappc9-deficient ASCs (Figure 5A–C), we wondered whether rab18-containing structures were similarly enlarged in trappc9-null ASCs under un-induced conditions. Fluorescent microscopic analysis revealed that rab18-positive signals occurred at punctate structures, which were distributed throughout the cytoplasm and appeared to form enclosed circular structures in some cases (Figure 5D). Quantitative analysis revealed that the size of rab18-labeled enclosed circular structures were enlarged in trappc9-null ASCs (Figure 5D,E). In addition, the intensities of rab18-immunoreactive signals in trappc9-deficient ASCs were increased related to those in WT ASCs (Figure 5F). Taken together, these data support the idea that trappc9 deficiency indeed alters the function of rab18 in ASCs.

### 2.5. The Content of Neural Stem Cells Declines in the Brain of Trappc9-Null Mice

As the volume of various brain structures in trappc9-deficient mice is reduced [18], we reasoned that the deficiency of trappc9 might also impinge neural stem or progenitor cells in the brain. To attest this idea, we first examined the expression level of Sox-2, a stem cell marker, in the brain of WT and trappc9-null mice. Both qRT-PCR and Western blot analyses showed that Sox-2 was expressed at lower levels in the brain of trappc9-null mice when compared with those of WT mice (Figure 6A–C). To determine whether the downregulated expression of Sox-2 reflected a bona fide decrease of the cell number, we processed a series of brain sections cut through the hippocampus for labeling with antibodies against Sox-2. Cell counting with stereology methods showed that the content of Sox-2-positive cells in both the subventricular zone of the lateral ventricle and the subgranular zone of the dentate gyrus was significantly reduced in trappc9-null mice relative to age-matched WT mice (Figure 6D–F). These findings suggest that the deficiency of trappc9 also decimates neural stem/progenitor cells in the brain.

## 3. Discussion

The functional compromise of stem cells accelerates the development of obesity and associated co-morbidity and considerable studies have been focused on the detrimental impacts of obesity on stem cells [5,6,7,8,38,39]. In this study, we examined whether the plasticity of stem cells was contracted in the presence of a genetic mutation of trappc9, which causes a global developmental delay with varied degrees of obesity in humans. We took advantage of newly generated trappc9-deficient mice, which grow normally during the weaning period but appear overweight rapidly after weaning [18]. We isolated stem or stromal cells from the abdominal adipose tissues of trappc9-null mice and WT littermates before overweight was developed and examined their proliferation and differentiation capacities. We found that trappc9 deficiency constricted the self-renewal of ASCs. While their lineage commitment was retained, trappc9-null ASCs manifested a bias toward adipogenic differentiation. On the cellular and molecular levels, trappc9-deficient ASCs had defects in coping with stress and in intracellular traffic engaging rab1, rab11, and/or rab18. Our study suggests that obesity induced by the deficiency of trappc9 involves the impaired plasticity of ASCs. Noticeably, trappc9 deficiency also impinges neural stem cells in the brain. This may be a basis of brain structure hypoplasia seen in patients with trappc9 mutations.

The failure of cytoplasmic division at the end of mitosis and the activation of programmed cell death are the reasons for the compromised self-renewal of the trappc9-deficient ASCs. Cytokinesis or cytoplasmic division into two cells requires a dynamic remodeling of local actin filaments and the addition of membranes to the growing furrow [40,41,42]. We noticed that there was a greater population of cells containing two or more nuclei in ASCs cultures obtained from trappc9-deficient mice than in ASC cultures from WT mice. A role for trappc9 in cytokinesis has been unveiled in a study, which shows that the loss-of-function of *Drosophila* trappc9, known as bru, halts cleavage furrow ingression during cytokinesis and abolishes rab11 to localize to the cleavage furrow [43]. Given that cytokinesis requires the proper function of rab11 [44,45], the failure of cytokinesis rendered by the deficiency of bru/trappc9 may arise from the compromised function of rab11. Indeed, rab11 in the cells of trappc9-null mice does not function properly, as indicated by the impeded recycling of the transferrin receptor in the primary neurons cultured from trappc9-deficient mice [18]. The abrogation of trappc9 in mice also alters the expression of kalirin, an activator for rac1, which plays a key role in actin cytoskeleton remodeling [46]. In this regard, the defective cytokinesis of trappc9-null ASCs may also involve the altered function of rac1.

How programmed cell death or apoptosis is activated in trappc9-deficient ASCs remains speculative. One possibility can be related to the failure of cytokinesis. Pohl and Jentsch have demonstrated that the depletion of BRUCE, a protein associated with rab11-positive endosomes and coordinating multiple events at the final stages of cytokinesis, arrests cell division and in the meanwhile induces marked cell death [47]. Alternatively, the increased apoptosis of trappc9-null ASCs may originate from the intracellular accumulation of ROS, which is known to be able to trigger apoptosis through various death-signaling pathways [48]. Compared with WT ASCs, trappc9-null ASCs contained higher levels of ROS. However, the mechanism that drives ROS accumulation in trappc9-null ASCs remains to be clarified. We propose that it may involve the improper function of rab11, as previous studies have found the improper function of rab11 in the cells of trappc9-deficient mice and have shown that the declined function of rab11 causes ROS accumulation [18,49]. Additionally, it is also possible that the overwhelming accumulation of lipid droplets in trappc9-deficient ASCs may increase the lipid metabolic flux in mitochondria to lead to excessive production of ROS [50].

Premature entry into the senescent state is another contributor to the poor proliferation of trappc9-null ASCs. Cellular senescence is a stress response that leads to the irreversible exit of the cell cycle and can be induced by diverse intrinsic and extrinsic stimuli [51]. How senescence is induced in trappc9-null ASCs remains to be defined, but may involve multiple factors. For an example, excessive ROS accumulated in trappc9-deficient ASCs might constantly damage DNA molecules and subsequently initiate cellular senescence [51]. A study reported that lipid balance was important to delaying the cellular senescence of human amniotic epithelial cells [52]. Hence, lipid metabolism disturbance seen in trappc9-null ASCs may contribute significantly to the premature cellular senescence. Autophagy is a fundamental cellular process by which aged organelles and other damaged macromolecules are degraded and plays a pivotal role in maintaining cellular homeostasis and protecting cells from senescence [26,31]. In senescing cells, autophagy activation is also needed for restricting further damage. We found that autophagy was enhanced, but autophagic degradation was impaired in trappc9-null ASCs, as reflected by the increased levels of autophagy substrates LC3-II and p62/SQSTM1. The decreased autophagic flux in trappc9-null ASCs was also supported by the accumulation of lipid droplets. Future studies are needed to dissect at which stages autophagic degradation is impaired and whether rab1, rab11, and/or rab18 are involved.

We noticed that there was an inconsistence between brain cells and ASCs regarding the effect of trappc9 deficiency on rab18. Compared with WT mice, trappc9-deficient mice express similar levels of rab18 in the brain [18]. However, in trappc9-deficient ASCs, levels of rab18 were vastly elevated relative to those in WT ASCs. This change is unlikely to be a non-specific effect because under the same conditions neither rab5 nor rab7 was altered in trappc9-null ASCs. We further employed an alternative approach to demonstrate that there was a bona fide impact on rab18 in trappc9-deficient ASCs. Consistent with our findings in the Western blot analysis, fluorescent microscopic cell imaging showed that levels of rab18-immunorective signals in trappc9-null ASCs were higher than those in WT ASCs. Furthermore, rab18-containing enclosed circular structures were enlarged in a population of trappc9-null ASCs. It will be interesting to know the factor(s) responsible for such an inconsistence.

In conclusion, we found that trappc9-deficient adipose stem cells were compromised in self-renewal and skewed toward the adipogenic differentiation. Such anomalies coincided with changes in multiple cellular pathways, particularly those engaging rab1, rab11, and/or rab18. We suggest that the development of overweight in trappc9-deficient mice involves the impaired plasticity of adipose stem cells. While we show the powerfulness of trappc9-deficient mice in investigating obesity development and uncover several cellular pathways that are perturbed by the deficiency of trappc9 in mice, our study is limited in exploring the molecular underpinnings behind the perturbation of these pathways and lacks attempts to examine whether the perturbation can be attenuated upon restoring trappc9. Future studies are also necessary to examine whether these changes occur in the stem cells of patients with trappc9 mutations.

## 4. Materials and Methods

### 4.1. Animals

Mice were housed in a pathogen-free environment with free access to water and food at a controlled temperature and with a standard 12 h light/dark cycle. All experimental procedures were approved by the Institutional Animal Care and Use Committee of the Shanghai Jiao Tong University (#A2017029) and carried out in accordance with NIH guidelines.

### 4.2. Isolation and Culture of Adipose-Derived Mesenchymal Stem Cells

Primary ASCs were isolated as previously described in references [53,54,55]. Briefly, adipose tissues were collected under aseptic conditions from abdominal fat pads of 3–4-week-old mice. For each of two independent preparations, adipose tissues from 3 mice for each genotype were pooled, washed three times with Hank’s balanced salt solution (HBSS), minced into pieces, and digested with 1 mg mL^−1^ collagenase type-1 (Sigma-Aldrich, St. Louis, MO, USA) for 40 min at 37 °C with gentle mixing. Enzymatic treatment was terminated by adding complete α-MEM media, which contained Minimum Essential Medium alpha basic (α-MEM, ThermoFisher, Suzhou, China), 10% fetal bovine serum (FBS, Gibco, Grand Island, NY, USA), 1× L-glutamine, and 1× penicillin/streptomycin (ThermoFisher, China). The collagenase digestion mixtures were then filtered through a 70 µm cell strainer and centrifuged for 10 min at 1, 200 rpm. The cell pellets and stromal vascular fractions (SVF) were re-suspended in complete α-MEM media and cultured at 37 °C in a humidified cell culture incubator (ThermoFisher Scientific, Waltham, MA, USA). Adherent cells, annotated as ASCs, were passaged and cultured as above. ASCs at passage-3 were analyzed by flow cytometry (BD LSR Fortessa, San Jose, CA, USA) with fluorescein isothiocynate (FITC)-conjugated antibodies for CD29 (Invitrogen, Waltham, MA, USA, #14-0292-82), CD44 (Invitrogen, #14-0441-82), Sca1 (Invitrogen, 11-5981-82), and CD45 (Invitrogen, #14-0451-82), respectively. Cells positive for the corresponding markers were quantified with the FlowJo software (version-7). After being verified, ASCs at passage-3 were collected and stored at −80 °C or liquid nitrogen until being used for studies.

### 4.3. Lineage Differentiation of ASCs

To examine the capacity of their differentiation to adipocyte and osteocyte lineages, ASCs at passage-3 were seeded on glass coverslips in a 6-well plate at 1 × 10^5^ cells per well. After being cultured in a humidified incubator for approximately 36 h, ASCs were changed into differentiation media and cultured until being processed for analysis. Fresh differentiation media were changed twice a week. Complete α-MEM media supplemented with 1 µM dexamethasone, 200 µM indomethacin, 0.5 mM Isobutyl-1-methylxanthine (Sigma-Aldrich), and 10 µg/mL human insulin (Yeasen Biotechnology, Shanghai, China) were used for adipogenic differentiation, whereas complete α-MEM media supplemented with 1 µM dexamethasone, 10 mM β-glycerophosphate, and 50 µg L^−1^ ascorbic acid bisphosphate were used for osteogenic differentiation. After induction for 14 days, ASCs on glass coverslips were fixed with 4% paraformaldehyde for 15 min, washed with PBS, and stained with 0.5% Oil red O (Sigma-Aldrich) or 1% Alizarin red (Sigma-Aldrich) for 15 min. After washes with PBS, cells on glass coverslips were mounted on glass slides for microscopy. Digital images were captured from randomly chosen fields using a bright-field Olympus BX53 microscope with the same settings.

### 4.4. Detection of Intracellular Reactive Oxygen Species

Intracellular reactive oxygen species (ROS) were measured using 2′,7′-dichlorofluorescein diacetate (DCFDA) (Beyotime Biotechnology, Shanghai, China) according to the supplier’s instructions. In brief, ASCs at passage-4 were seeded in a 24-well plate and cultured until approximately 90% confluency. After one wash with PBS, ASCs were incubated with 10 µmol mL^−1^ DCFH-DA for 30 min at 37 °C in dark, washed several times with PBS, and then imaged using an Olympus IX73 inverted-fluorescence microscope. Digital images were collected from randomly chosen fields with the same parameters for each genotype.

### 4.5. Detection of Senescence-Associated β-Galactosidase Activity

Senescence-associated β-galactosidase (SA-β-gal) was examined with a β-galactosidase staining kit (Beyotime Biotechnology, China) according to the manufacturer’s instructions. Briefly, ASCs were seeded on glass coverslips in a 6-well plate at 1 × 10^5^ cells per well and cultured for approximately 36 h. After washes with PBS, ASCs on glass coverslips were fixed with 4% paraformaldehyde for 15 min and incubated with β-gal staining solution overnight in a 37 °C incubator. Cells on glass coverslips were then washed several times with PBS and mounted on glass slides. Cells were examined under a bright-field Olympus BX53 microscope. Digital images were captured from randomly selected areas with the same imaging parameters. Cells positive for β-gal were counted with the NIH ImageJ software 1.48v.

### 4.6. Quantitative Real-Time PCR

Total RNAs were isolated using Trizol reagents (Ambion, Austin, TX, USA). First strand cDNAs were synthesized using a Hifair II 1st Strand cDNA Synthesis SuperMix for qPCR (Yeasen, Shanghai, China, Cat:1123ES60). Real-time PCR was performed with TB Green Premix Ex Taq (Takara Bio Inc., Kusatsu, Japan) and Step One Real-Time PCR system (Applied Biosystems 7300, Foster City, CA, USA). Primers used for qPCR included: 5′ tgtgatgcctttgtgggaacc and 5′ cgtctgcggtgatttcatc (*Ap2*), 5′ aggtcatcttctgtgctagg and 5′ atgctggaagacctgctatg (*Lpl*), 5′ gggtcagctcttgtgaatgg and 5′ ctgatgcactgcctatgagc (*Ppar-γ*), 5′ cagaggaggcaagcgtcact and 5′ ctgtctgggtgccaacactg (*Bsp*), 5′ aacccacggccctccctgaactct and 5′ actggcggggtgtaggtaaaggtg (*Runx2*), 5′ accctggctgcgctctgtctct and 5′ gatgcgtttgtaggcggtcttca (*Oc*), 5′-aacatctcagggccgaaa and 5′-tgcgcttggagtgatagaaa (*p21*), 5′ gtgtgcatgacgtgcggg and 5′ gcagttcgaatctgcaccgtag (*p16*), 5′ tggaggagtcacagtcggat and 5′ cgtccatgcagtgaggtgat (*p53*), 5′ gcggagtggaaacttttgtcc and 5′ cgggaagcgtgtacttatcctt (*Sox-2*), and 5′ ctaaggccaaccgtgaaaag and 5′ accagaggcatacagggaca (*β-Actin*). PCR was performed in triplicate for each gene. The gene expression level was determined using the 2^-ΔΔCt^ method, with β-Actin as a normalization control.

### 4.7. SDS-PAGE and Western Blotting

Protein lysates were prepared from cultured cells or brain tissues using RIPA buffers containing protease inhibitors (Biotool B14011, Bimake, Houston, TX, USA). The concentration of proteins was determined with the BCA kit (Beyotime Biotechnology, China). SDS-PAGE and Western blot were conducted as standard procedures. Primary antibodies used for Western blot analysis included: rabbit anti-rab1A (1:1000, Cell Signaling, D5F8M), rabbit anti-rab5 (1:500, PA5-29022, Invitrogen/ThermoFisher), rabbit anti-rab7 (1:500, PA5-23138 Invitrogen, Waltham, MA, USA), mouse anti-rab11 (1:1000, AB_397984, BD Transduction Laboratories, Franklin Lakes, NJ, USA ), mouse anti-rab18 (1:500, sc-393168, Santa Cruz Biotechnology, Dallas, TX, USA), rabbit anti-LC3 (1:1000, 14600-1-AP, ProteinTech, Wuhan, China), rabbit anti-p62 (1:1000, 18420-1-AP, ProteinTech), rabbit anti-Beclin-1 (1:1000, 3613, ProSci, Fort Collin, CO, USA), mouse anti-Sox2 (1:1000, AB_5603, Merck Millipore, Burlington, MA, USA), mouse anti-ACTIN (1:5000, 66009-1-Ig, ProteinTech), and mouse anti-GAPDH (1:5000, 60004-1-Ig, ProteinTech). Peroxidase AffiniPure goat anti-mouse IgG (H + L) (115035003) and anti-rabbit IgG (H + L) (111035003) (Jackson immune research, West grove, USA) were used as a dilution of 1:5000, Protein signals were detected with ECL reagents (Yeasen Biotechnology, China) and imaged with the ChemiDoc MP imaging system (Bio-Rad, Hercules, CA, USA). Signal intensities were measured using the NIH ImageJ software 1.48v.

### 4.8. Immunofluorescence Microscopy

Mice at the age of 3–4 weeks were anesthetized and intracardially perfused with PBS followed by freshly prepared 4% paraformaldehyde. Brain tissues were collected and post-fixed in 4% PFA overnight at 4 °C. Fixed brain tissues were dehydrated by being sequentially soaked in 10, 20, and 30% sucrose overnight at 4 °C, and then embedded in OCT media (Leica Microsystem, Mannheim, Germany). Coronal brain sections were cut at −30 °C with an interval of 30 µm on a cryostat (Leica CM1950). A series of consecutive brain sections cut through the hippocampus were processed for labeling with antibodies against Sox-2 (1:100, AB_5603, Merck Millipore), as described in previous studies [18]. Cells in brain sections were identified by staining nuclei with Hoechst 33258 (1:2000, H3569, Invitrogen). Labeled brain sections were mounted on glass slides for microscopy. Digital images were acquired with the Olympus BX53 fluorescence imaging system with the same settings.

To examine rab18-containing structures, ASCs at passage-4 were cultured on glass coverslips in a 12-well plate overnight, washed with PBS, and fixed in 4% PFA for 15 min. Cells on glass coverslips were incubated with 0.1 M Tris/Cl (pH7.4) to quench free aldehyde groups. After being blocked with 2% bovine serum albumin (BSA), cells were incubated with anti-rab18 antibodies (1:100, sc-393168, Santa Cruz Biotechnology) at 4 °C overnight followed by washes with PBS and incubation with Alexa-488-conjugated goat anti-mouse secondary antibody (1:1000, Abcam, Cambridge, UK). Before being mounted onto glass slides, cells were stained with Hoechst 33258. Digital images were acquired through a 63× oil immersion objective using the Leica TCS SP8 confocal imaging system with the same settings, including laser strength, pinhole, signal gain, and resolution.

### 4.9. Image Analysis

All images were analyzed using NIH ImageJ software by at least two examiners who were blinded to experimental conditions. The number of nuclei per cell in each image was counted using the cell-counter plugin of NIH ImageJ. For measuring intensities of Oil red, Alizarin red, DCF, and/or rab18-immunoreactive signals, images were converted to 8-bit black-and-white ones. Each cell in the image was identified by manually tracing the cell edge using the free-hand tool of NIH ImageJ. Signal intensities within each contour were measured and the corresponding background signals were removed. For measuring the size of rab18-decorated circular structures in a cell, contours were drawn around each enclosed structure using the free-hand tool and the enclosed areas were measured.

To count Sox-2-positive cells in the subventricular zone (SVZ) and subgranular zone (SGZ) of immune-labeled brain sections, images obtained from a series of 3 consecutive brain sections from each of 3 WT and 3 trappc9-null mice were used for stereology quantification, as in our previous studies [18]. In brief, images of blue and red channels were merged. The SVZ or SGZ in each merged image was demarcated with a contour using the free-hand tool of NIH ImageJ. The number of Sox-2-positive cells within the contour was determined with the “measure” function of NIH ImageJ. The readout of areas within the contours was converted into mm^2^ based on the resolution of images. The accuracy of cell counting was further evaluated by manually verifying the co-localization between Sox-2-immunoreactive and Hoechst 33258 signals. Data were represented as cell number per cross-sectional area.

### 4.10. Statistical Analysis

All data were presented as Mean ± SD. An unpaired two-tailed Student’s *t*-test was used to determine the significance between the mean of two groups. A *p* value less than 0.05 was considered statistically significant.

## Figures and Tables

**Figure 1 ijms-23-04900-f001:**
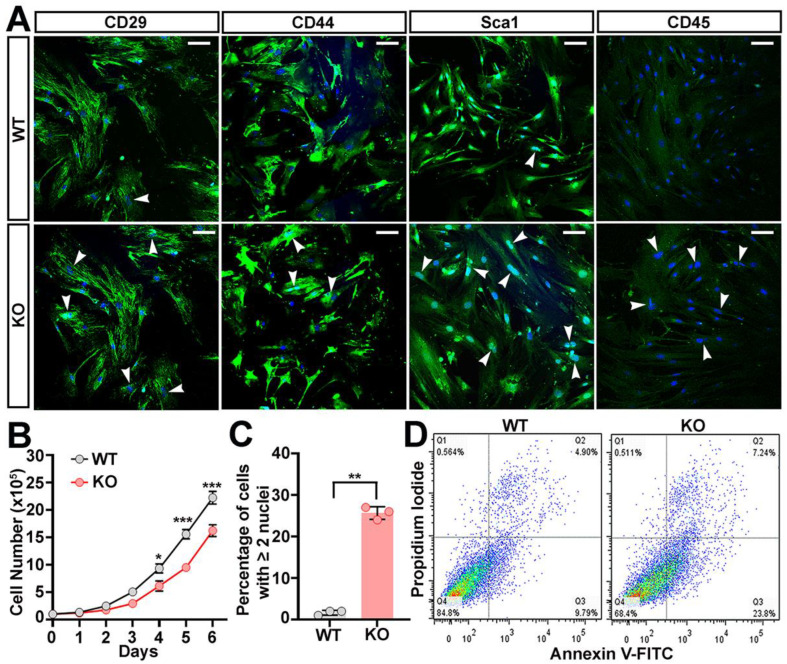
Compromised self-renewal of ASCs cultured from trappc9-null mice. ASCs were isolated from WT and trappc9-deficient mice and cultured as in Methods. Both WT and KO ASCs at passage-3 were subjected to analyses. (**A**) Confocal images showed that ASCs isolated from WT and trappc9-null (KO) mice expressed mesenchymal stem cell markers and were devoid of hematopoietic CD45. Arrowheads point to cells containing two or more nuclei. Scale bars: 100 μm. (**B**) Comparison of the growth rate between WT and trappc9-deficient (KO) ASCs. ASCs were cultured at a density of 1 × 10^5^ in T25 flasks and counted every 24 h for 6 days. Cell counting was performed with a hemocytometer. (**C**) Quantification of cells with two or more nuclei revealed that trappc9-null (KO) ASCs were defective in cell division. Images captured from 3 randomly chosen visual fields from cells for each genotype were used for cell counting. (**D**) Apoptosis of cells in WT and trappc9-deficient (KO) ASC cultures was examined by flow cytometry-based analysis of phosphatidylserine exposure (Annexin V-FITC staining) and plasma membrane integrity (propidium iodide staining). Shown are dot plot graphs from one of three individual analyses. Similar observations were achieved in other two experiments. Data in (**B**,**C**) are Mean ± SD. Each symbol represents one experiment. Statistical significance was determined by unpaired two-tailed Student’s *t*-test: * *p* < 0.05; ** *p* < 0.01; *** *p* < 0.001.

**Figure 2 ijms-23-04900-f002:**
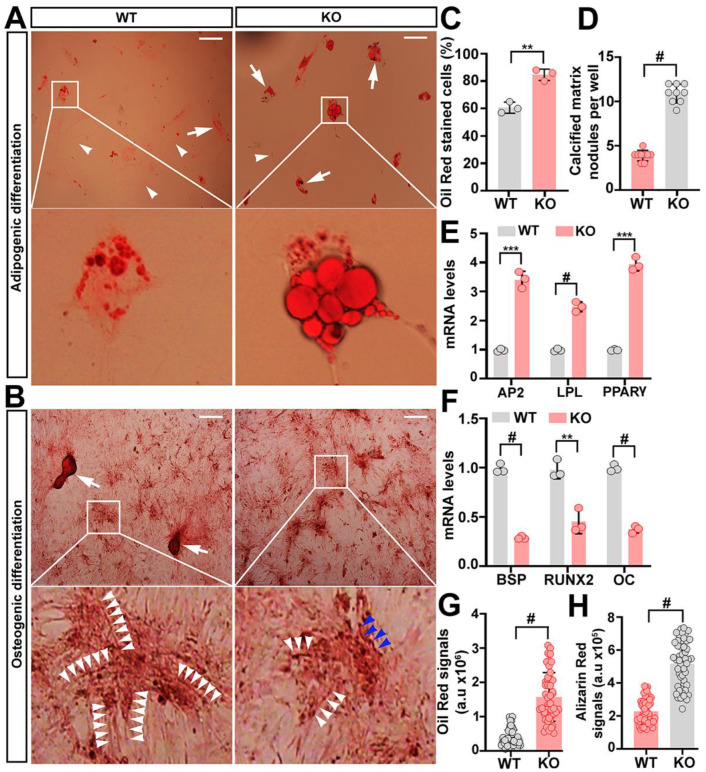
(**A**–**H**) Comparison of the lineage differentiation between WT and trappc9-deficient ASCs. After induced differentiation to the adipogenic (**A**) or osteogenic (**B**) lineage for 14 days, cells were stained with Oil red O (**A**) or Alizarin red (**B**) and imaged. Shown are examples of images of cells stained with Oil red O (**A**) or Alizarin red (**B**). Boxed regions were enlarged and shown beneath the corresponding image. Enlarged boxed regions in (**A**) highlighted clusters of aberrantly large lipid droplets in an adipogenic cell derived from trappc9-null (KO) ASCs. In images in (**A**), arrowheads point to cells that were stained with oil red, whereas arrows identified cells stained with oil red. Arrowheads in images of the enlarged boxed regions in (**B**) indicated the string pattern of mineral deposition in osteoblasts derived from WT ASCs; such string-like structures were twisted (blue arrowheads) in osteoblasts from trappc9-null (KO) ASCs. Scale bars: 100 μm. Quantitative analysis of cells stained with Oil red or with Alizarin red (**C**) as well as qPCR analysis of transcripts for adipogenic (**D**) and osteogenic (**E**) genes revealed that trappc9-deficient (KO) ASCs preferred adipogenic differentiation. In (**C**), average percentage of cells stained with oil red was graphed, whereas average calcified matrix nodules per well was counted and used as a measure of osteogenic differentiation. (**F**) Densitometry quantification of signals for oil red (**A**) or Alizarin red (**B**) showed that upon induction, cells derived from trappc9-null (KO) ASCs enriched more lipids and deposited minerals to a lesser degree than cells derived from WT ASCs. All experiments were repeated three times. Data are Mean ± SD. Unpaired two-tailed Student’s *t*-test: ** *p* < 0.01, *** *p* < 0.001, # *p* < 0.0001.

**Figure 3 ijms-23-04900-f003:**
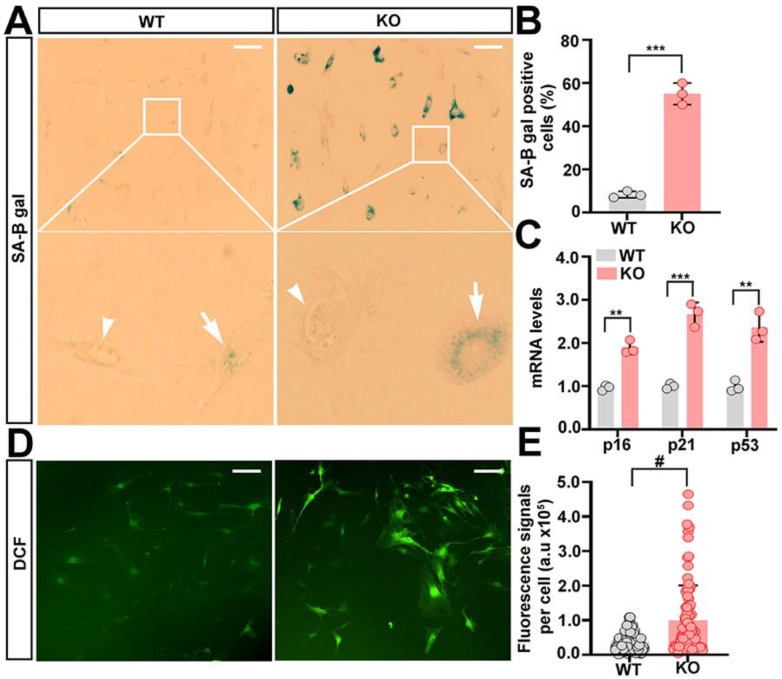
Trappc9-deficient ASCs are under increased stress. WT and trappc9-null ASCs at passage-4 were processed for analyzing cellular senescence (**A**–**C**) and measuring levels of reactive oxygen species (**D**,**E**). Microscopic analysis of ASCs stained for detecting SA-β-gal activity (**A**), followed by quantitative analysis of cells positive for SA-β-gal (**B**), and qPCR analysis of expression levels of senescence-related genes (**C**) showed that compared with WT ASCs, trappc9-deficient (KO) ASCs at early passages were prone to cellular senescence. Boxed regions were enlarged and shown beneath the corresponding image. Cells negative and positive for SA-β-gal were identified by arrowheads and arrows, respectively. (**D**,**E**) Accumulation of ROS in trappc9-deficient ASCs. Upon exposure to non-fluorescent DCFDA, which is converted to fluorescent DCF in the presence of reactive oxygen species, both WT and trappc9-deficient (KO) ASCs exhibited green fluorescence signals, as indicated in confocal images (**D**). Densitometry of DCF fluorescence signals revealed higher levels of reactive oxygen species in trappc9-deficient (KO) ASCs than in WT ASCs (**E**). Scale bars: 100 μm. Data are Mean ± SD. Each symbol represents one experiment in (**B**,**C**) and one cell in (**E**). Unpaired two-tailed Student’s *t*-test: ** *p* < 0.01, *** *p* < 0.001, # *p* < 0.0001.

**Figure 4 ijms-23-04900-f004:**
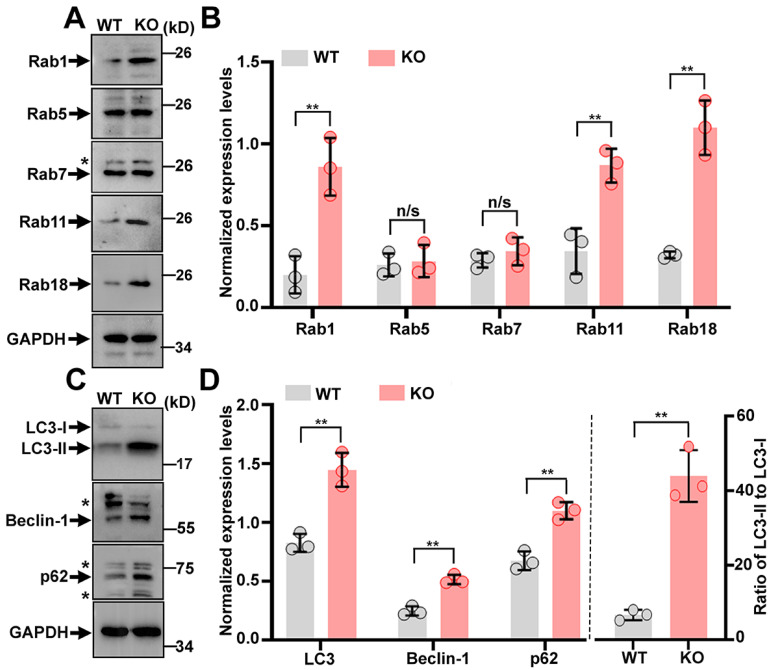
Trappc9 loss impacts the expression of rab proteins and autophagy in ASCs. Post-nuclear supernatants prepared from WT and trappc9-deficient (KO) ASCs at passage-4 were subjected to SDS-PAGE and Western blot analysis with antibodies for indicated rab proteins (**A**,**B**) and proteins related to autophagy (**C**,**D**). GAPDH was a loading control. Blot analyses shown in (**A**,**C**) were from one of three independent experiments. Densitometry quantification of signal intensities for each of the indicated proteins were graphed and shown in (**C**,**D**), respectively. Expression levels of indicated proteins were normalized with signal intensities of the corresponding GAPDH. Star symbols beside gel graphs indicated proteins cross-reactive to the corresponding antibodies. Data are Mean ± SD. Each symbol in (**B**,**D**) represents one experiment. Unpaired two-tailed Student’s *t*-test: * *p* <0.05; ** *p* < 0.01; n/s, no significance.

**Figure 5 ijms-23-04900-f005:**
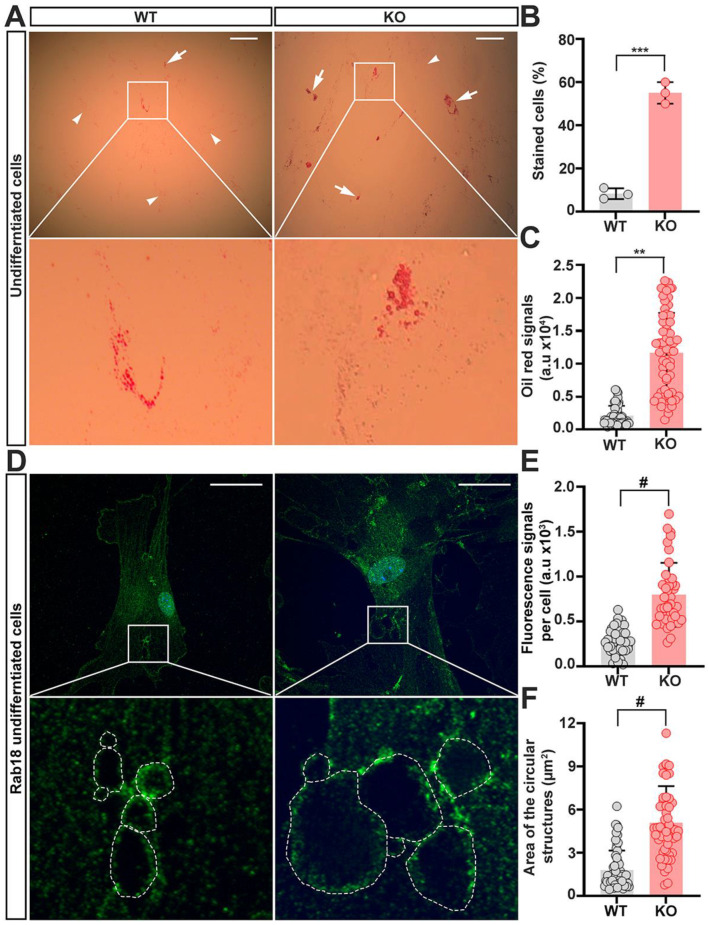
Enlargement of lipid droplets occurs in un-induced trappc9-deficient ASCs. WT and trappc9-deficient ASCs at passage-4 were seeded in 6-well plates with or without glass coverslips and cultured overnight. Cells in wells without glass coverslips were stained with oil red, whereas cells on glass coverslips were processed for immuno-labeling rab18. (**A**) Representative images of oil red stained cells. Enlarged boxed regions beneath the corresponding image showed clusters of enlarged lipid droplets in un-induced trappc9-null (KO) ASCs. Arrows identified cells having clustered large lipid droplets, whereas arrowheads indicated cells lacking such clustered lipid droplets. Scale bars: 100 μm. (**B**) Cell counting revealed more trappc9-defidient (KO) ASCs displaying clustered lipid droplets than WT ASCs. For each of 3 independent experiments, 100 cells were counted from 3 images taken from randomly chosen and non-overlapped visual fields. (**C**) Densitometry of oil red signals suggested greater accumulation of lipids in trappc9-null (KO) ASCs than in WT ASCs. (**D**) Representative confocal images of cells labeled with antibodies for rab18. Dashed contours in enlarged boxed regions beneath the corresponding image indicated circular structures formed by rab18-positive signals. Scale bars: 10 μm. (**E**) Densitometry of rab18-immunoreactive signals revealed upregulation of rab18 in trappc9-deficient (KO) ASCs. (**F**) Measurement of the size of circular structures in cells as highlighted in images (**D**), showed enlargement of rab18-positive structures in trappc9-deficient (KO) ASCs. The size of rab18 decorated circular structures from 15 cells for each genotype was measured. Data are Mean ± SD. Each symbol represents one experiment in (**B**), one cell in (**C**,**E**), and one circular structure in (**F**). Unpaired two-tailed Student’s *t*-test: ** *p* < 0.01; *** *p* < 0.001; # *p* < 0.0001.

**Figure 6 ijms-23-04900-f006:**
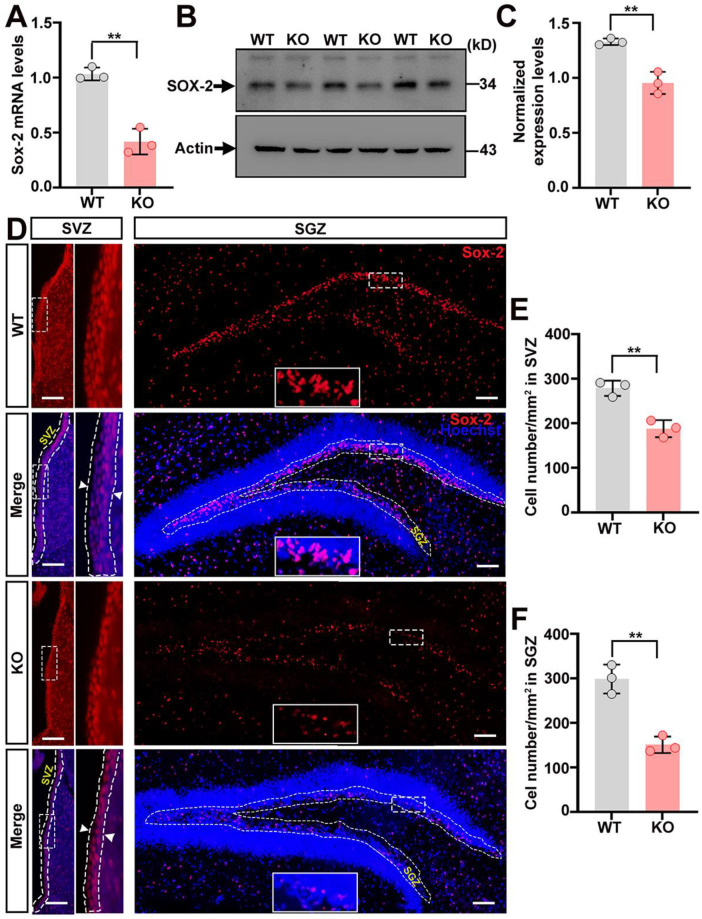
The content of neural stem cells declines in the brain of trappc9-deficient mice. Quantitative PCR (**A**) as well as Western blot (**B**) followed densitometry (**C**) analyses showed lower expression levels of Sox-2 in the brain of 3-week-old trappc9-null (KO) mice as compared to age-matched WT mice. Actin was a reference or loading control. Expression levels of Sox-2 in the brain were normalized with signal intensities of actin. (**D**) A series of 3 coronal brain sections of 3-week-old WT and trappc9-null (KO) mice were processed for immuno-labeling Sox-2 (red) and staining with Hoechst 34580 (blue) to identify all cells in brain sections. Images for red and blue channels were captured separately through a 20× objective with the Olympus BX53 imaging system. Merged images of different areas were re-aligned into one image of that brain section. Dashed contours indicated the subventricular zone (SVZ) and subgranular zone (SGZ). Boxed regions were enlarged and shown to the right (SVZ) or in the lower-middle region (SGZ) of the corresponding image. Arrowheads in the enlarged boxed regions indicated the thickness of the SVZ zone. Scale bar: 100 μm. (**E**,**F**) Cell counting of Sox-2-positive cells within dashed SVZ and SGZ contours showed a decreased number of cells in trappc9-null (KO) mice as compared to WT mice. Data are Mean ± SD. Two tailed Student’s *t*-test: ** *p* < 0.01.

## Data Availability

All data needed to evaluate the conclusions in the paper are present in the paper and/or the Appendix A. Additional data related to this paper may be requested from the authors.

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
