# Peer review of "Trappc9 Deficiency Impairs the Plasticity of Stem Cells"

_ijms, 2022, doi:10.3390/ijms23094900_

Round 1
Reviewer 1 Report
This is a well written manuscript titled "Trappc9 deficiency expedites cellular senescence and skews the lineage differentiation of adipose-derived mesenchymal stem cells". My suggestions are as below:
- Suggest that authors identify study limitations (if any).
- Suggest authors describe the clinical relevance/implications of the study findings.
- Suggest authors outline some future research directions.
Author Response
This is a well written manuscript titled "Trappc9 deficiency expedites cellular senescence and skews the lineage differentiation of adipose-derived mesenchymal stem cells". My suggestions are as below:
1. Suggest that authors identify study limitations (if any).
2. Suggest authors describe the clinical relevance/implications of the study findings.
3. Suggest authors outline some future research directions.
We appreciate this review for the enthusiasm to our work and many thanks for suggestions.
We have noted future studies in relevant paragraphs in the Discussion section. As suggested, we further discussed the limitations of our studies and future research directions in the last paragraph of the Discussion section.
Reviewer 2 Report
Broad comments
I honestly appreciated the research plan and how the experiments and results have been represented. I have some comments to improve the manuscript that, overall, is good quality. The paper is interesting for obesity research and MSC experts in pathology and physiology studies. Materials and methods are well described. The data about adipo/osteo differentiation fate, ROS levels and premature senescence with relevance to obesity and SOX2 in neural cells were been considered interesting to me.
Major comments
- Alizarin Red used in Figure 2, by forming complex, stains not “minerals”, but calcium deposits. Please be more precise in the figure legend, as well as in abstract and section “results”, thanks. Th authors indeed expressed better idea at line 128-131.
- About Figure 2 C and F, I suggest to separate oli red data and alizarin red data, thus creating two new graphs by coupling for the single staining (e.g. oil red stained cells % + quantification signal). This will work better also by seeing the qPCR results that instead are already clearly separated.
- Authors wrote that “two or more nuclei in trappc9-null ASC cultures than in WT ASC cultures 82 (Figure 1C), suggestive of impaired cell division or cytokinesis. “ . I suggest to remove the hypothesis after the comma, because this should be further investigated and it’s a vague information.
- I recommend to highlight the importance of lipid balance also in vitro for MSC culture and other cells such as stem cells (e.g. from placenta) in influencing the senescence level (please have a look at the paper doi.org/10.3390/jcm11051236). In this context, do the authors want to briefly mention and discuss this aspect for ASC properties? For example some questions are:
a) can be how balance of lipid and lipid profile KO ASCs?
b) which role may have trappc9 during in vitro expansion of in vitro cultured human ASC (e-g- proliferation or anti-senescence role)?
c) What about overexpression of trappc9 in mouse stem cells (ASCs or NSC? others? )
d) Can this compromise normal function of the stem cells? May be feasible to overexpress such gene also in human ASC? Any preliminary investigation? thanks - Figure 1 C axis is “Percent”, but would “percentage” be better?
- Would be nice to add at least the staining of non-induced / basal/ undifferentiated ASC and KO-ASC for both oil red and alizarin in the supplementary data. About this, is there any spontaneous difference and partial differentiation between wild type and knock out not induced with osteogenic medium, but cultured for many days in parallel to osteogenic induced samples?
- Since in figure 6 legend it is written that “Actin was a reference or loading control. “I wonder if in Figure 6C is really a ratio or a normalization for actin signal?
- Please highlight in the concluding part of the discussion section n°3 both the limitations of the study (ASCs are from mice and not human, thus not clinically relevant) and again remind its advantages (animal model for MSC role in genetics of the obesity)
- the title and the majority of the manuscript focuses on the ASC, on the other hand authors included data avout trappc9 deficiency which also impinges neural stem cells. While I can understand that results are not enough to make two different papers, maybe a different title can be arranged. For example, “effects of trappc9 deficiency on the plasticity of mouse adipose-derived mesenchymal stem cells and neural stem cells.” Or othe titles more appropriate thought by the authors.
Minor comments
- Fig 1 D legend., typo “trappc9-deficien “, it misses the “t”
- Line 503 typo, Alizerian
- On Line 126 disrupts , I suggest “disturbs”
Finally, just a kind reminder. Remove supplementary file at the proof level
Author Response
We appreciated this reviewer for the interest and thoughtful comments and suggestions. We addressed the comments in a point-by-point manner as below.
Major comments
- Alizarin Red used in Figure 2, by forming complex, stains not “minerals”, but calcium deposits. Please be more precise in the figure legend, as well as in abstract and section “results”, thanks. Th authors indeed expressed better idea at line 128-131.
As suggested, we changed “minerals” into “calcium or calcium deposits” throughout the manuscript.
- About Figure 2 C and F, I suggest to separate oli red data and alizarin red data, thus creating two new graphs by coupling for the single staining (e.g. oil red stained cells % + quantification signal). This will work better also by seeing the qPCR results that instead are already clearly separated.
We have split the original Figure 2C into new 2C and 2D, and original 2F into new 2G and 2H.
- Authors wrote that “two or more nuclei in trappc9-null ASC cultures than in WT ASC cultures 82 (Figure 1C), suggestive of impaired cell division or cytokinesis. “ . I suggest to remove the hypothesis after the comma, because this should be further investigated and it’s a vague information.
As suggested, we deleted the statement “suggestive of impaired cell division or cytokinesis”.
- I recommend to highlight the importance of lipid balance also in vitro for MSC culture and other cells such as stem cells (e.g. from placenta) in influencing the senescence level (please have a look at the paper doi.org/10.3390/jcm11051236). In this context, do the authors want to briefly mention and discuss this aspect for ASC properties? For example some questions are:
a) can be how balance of lipid and lipid profile KO ASCs?
b) which role may have trappc9 during in vitro expansion of in vitro cultured human ASC (e-g- proliferation or anti-senescence role)?
c) What about overexpression of trappc9 in mouse stem cells (ASCs or NSC? others? )
d) Can this compromise normal function of the stem cells? May be feasible to overexpress such gene also in human ASC? Any preliminary investigation? thanks
We emphasized the contribution of lipid balance to cellular senescence of trappc9-null ASCs by adding two sentences “A very study reported that lipid balance was important to delaying cellular senescence of human amniotic epithelial cells [52]. Hence, lipid metabolism disturbance seen in trappc9-null ASCs may contribute significantly to the premature cellular senescence.” and cited the reference mentioned by this reviewer.
However, the current lockdown in Shanghai because of COVID-19 pandemic prohibits us from any access to our laboratory to conduct any experiment.
5. Figure 1 C axis is “Percent”, but would “percentage” be better?
As suggested, we changed “percent” into “percentage”.
- Would be nice to add at least the staining of non-induced / basal/ undifferentiated ASC and KO-ASC for both oil red and alizarin in the supplementary data. About this, is there any spontaneous difference and partial differentiation between wild type and knock out not induced with osteogenic medium, but cultured for many days in parallel to osteogenic induced samples?
The current lockdown in Shanghai because of COVID-19 pandemic prohibits us from any access to our laboratory to conduct any experiment including the experiments suggested by this reviewer.
- Since in figure 6 legend it is written that “Actin was a reference or loading control. “I wonder if in Figure 6C is really a ratio or a normalization for actin signal?
We used shorter-exposure images of actin signals for quantification. Based on these conditions, the ratio was used for graphing the data. While we changed the actin photograph in the original Figure 6C into the shorter-exposure one used for densitometry, we also changed the labeling of the Y-axis from “Expression levels (ratio to actin)” into “Normalized expression levels”. We noted this change in the figure legend as ” Expression levels of Sox-2 in the brain were normalized with signal intensities of actin.”.
Please note, we also changed the labeling of the Y-axis in Figure 4 from “Expression levels (ratio to GAPDH)” into “Normalized expression levels” for the same reason. We noted this change in the figure legend as “Expression levels of indicated proteins were normalized with signal intensities of the corresponding GAPDH.”.
- Please highlight in the concluding part of the discussion section n°3 both the limitations of the study (ASCs are from mice and not human, thus not clinically relevant) and again remind its advantages (animal model for MSC role in genetics of the obesity)
As suggested, we discussed the limitations and advantages in the last paragraph of the Discussion section.
- the title and the majority of the manuscript focuses on the ASC, on the other hand authors included data avout trappc9 deficiency which also impinges neural stem cells. While I can understand that results are not enough to make two different papers, maybe a different title can be arranged. For example, “effects of trappc9 deficiency on the plasticity of mouse adipose-derived mesenchymal stem cells and neural stem cells.” Or the titles more appropriate thought by the authors.
As suggested, we changed the title into “Trappc9 deficiency impairs the plasticity of stem cells”.
Minor comments
- Fig 1 D legend., typo “trappc9-deficien “, it misses the “t”
We corrected this error.
- Line 503 typo, Alizerian
We corrected this error.
- On Line 126 disrupts, I suggest “disturbs”
As suggested we changed “disrupts” into “disturbs”.
Finally, just a kind reminder. Remove supplementary file at the proof level
Many thanks once again.